# Development of SSR Markers and Evaluation of Genetic Diversity of Endangered Plant *Saussurea involucrata*

**DOI:** 10.3390/biom14081010

**Published:** 2024-08-15

**Authors:** Lin Hu, Jiancheng Wang, Xiyong Wang, Daoyuan Zhang, Yanxia Sun, Ting Lu, Wei Shi

**Affiliations:** 1College of Forestry and Landscape Architecture, Xinjiang Agricultural University, Urumqi 830011, China; hulin54@163.com; 2State Key Laboratory of Desert and Oasis Ecology, Key Laboratory of Ecological Safety and Sustainable Development in Arid Lands, Xinjiang Institute of Ecology and Geography, Chinese Academy of Sciences, Urumqi 830011, China; www-1256@ms.xjb.ac.cn (J.W.); wangxy@ms.xjb.ac.cn (X.W.); zhangdy@ms.xjb.ac.cn (D.Z.); 3Xinjiang Key Lab of Conservation and Utilization of Plant Gene Resources, Xinjiang Institute of Ecology and Geography, Chinese Academy of Sciences, Urumqi 830011, China; 4Turpan Eremophytes Botanical Garden, Chinese Academy of Sciences, Turpan 838008, China; 5Center of Conservation Biology, Core Botanical Gardens, Chinese Academy of Sciences, Wuhan 430074, China; sunyanxia@wbgcas.cn

**Keywords:** genetic diversity, SSR maker, microsatellites, conservation, *Saussurea involucrata*

## Abstract

The conservation biology field underscores the importance of understanding genetic diversity and gene flow within plant populations and the factors that influence them. This study employs Simple Sequence Repeat (SSR) molecular markers to investigate the genetic diversity of the endangered plant species *Saussurea involucrata*, offering a theoretical foundation for its conservation efforts. Utilizing sequencing results to screen SSR loci, we designed and scrutinized 18 polymorphic microsatellite primers across 112 samples from 11 populations in the Bayinbuluke region. Our findings reveal high genetic diversity (I = 0.837, He = 0.470) and substantial gene flow (Nm = 1.390) among *S. involucrata* populations (China, Xinjiang), potentially attributed to efficient pollen and seed dispersal mechanisms. Principal Coordinate Analysis (PCoA) indicates a lack of distinct genetic structuring within the Bayinbuluke populations. The cluster analysis using STRUCTURE reflected the genetic structure of *S. involucrata* to a certain extent compared with PCoA. The results showed that all samples were divided into four groups. To safeguard this species, we advocate for the in situ conservation of all *S. involucrata* populations in the area. The SSR markers developed in this study provide a valuable resource for future genetic research on *S. involucrata*.

## 1. Introduction

Biodiversity encapsulates the variety of life forms on Earth, encompassing a spectrum of plants, animals, microorganisms, the genetic codes they harbor, and the ecosystems they forge. It is a multifaceted concept that includes ecosystem diversity, species diversity, and genetic diversity [1]. Among these, genetic diversity stands as a pivotal attribute, underpinning the capacity of species to adapt to environmental shifts and serving as the cornerstone of evolutionary processes [2,3]. Preserving genetic diversity is integral to formulating effective conservation and management strategies [4]. The extent of genetic variation and the population genetic structure of plant species are shaped by a tapestry of factors, including the evolutionary history, distribution range, morphology, reproductive strategies, and seed dispersal mechanisms. These factors are instrumental in crafting strategies for safeguarding species’ genetic variability [5].

*Saussurea involucrata* (Kar. & Kir.) Sch. Bip (Asteraceae), a perennial flowering and fruiting plant, thrives in alpine environments, often found in slopes, valleys, meadows, and rock crevices at elevations ranging from 2400 to 4100 m [6]. Predominantly native to Xinjiang, China, this species completes its life cycle from germination to flowering and fruiting in approximately six years. Peak blooming occurs from July to August, with a flowering duration of about four months. The unique spatial arrangement of male pre-maturation and hermaphroditic flowers facilitates crosspollination. The plant’s umbrella-shaped inflorescence and the manner in which its flowers open render it well suited to the harsh alpine climate. The extended period of pollination and nectar secretion increases the chances of successful pollination, leading to the production of a substantial number of seeds. The wind-dispersed achenes, equipped with a pappus, enhance seed germination and seedling survival under favorable conditions [7,8,9]. Recognized in traditional Chinese medicine, *S. involucrata* has been utilized ethno-medically for its diverse pharmacological properties, including anti-inflammatory, antifatigue, radiation-preventive, antitumor, free radical-scavenging, and anti-aging effects [10,11,12]. However, the natural reproduction rate of *S. involucrata* is inherently low, and its growth is slow [13,14]. The exploitation of its medicinal properties has led to excessive harvesting during its flowering period, impeding achene formation and causing a drastic decline in its natural reserves [15]. The Chinese Species Red List has classified *S. involucrata* as VU grade (http://protection.especies.cn/chineseredlist/list (accessed on 2 April 2024)), and it has been included in the national protected plant list as a secondarily protected species (http://www.iplant.cn/bhzw/info/1102 (accessed on 2 April 2024)).

The conservation genetics of *S. involucrata* in the Tianshan Mountains of Xinjiang have revealed high genetic diversity, with the western Tianshan Mountains and particularly the Bayinbuluke area emerging as genetic differentiation centers for the species [16,17]. The Bayinbuluke area, recognized as one of Central Asia’s biodiversity hotspots [18], offers an exemplary setting for studying the origins and conservation of biodiversity.

SSRs, or simple sequence repeats, were initially discovered by Tautz and Renz [19] and later termed microsatellites by Litt and Luty [20]. These short tandem repetitive sequences of 1–6 nucleotides are pervasively present across the genomes of eukaryotes [21]. Characterized by high polymorphism, repeatability, codominant expression, and neutrality, SSRs serve as invaluable tools for elucidating population genetic structures, gene flow, genetic relationships, and population viability. They are instrumental in assessing the impacts of habitat fragmentation and in guiding conservation strategies [22,23].

This study employs SSR markers to assess the genetic diversity of *S. involucrata* in the genetic differentiation center of the Bayinbuluke area, aiming to provide a scientific basis for the species’ conservation and sustainable management.

## 2. Materials and Methods

### 2.1. Plant Sampling and Preservation

In this study, we sampled 11 wild populations of *S. involucrata* in the Bayinbuluke area; each population was at least 7 km apart. In each population, healthy individuals of *S. involucrata* were randomly sampled (each individual was at least 50 m apart), and their leaves were collected and dried in silica gel. A total of 112 individuals were sampled from 11 populations. In the process of collection, the latitude and longitude of each sampling population were recorded. The location information of the *S. involucrata* sampling population and the sampling individuals of each population are shown in Figure 1.

### 2.2. Identification and Development of Genomic SSRs

In this study, the whole genome sequencing results of *S. involucrata* obtained from NCBI (project registration number: PRJNA991078) were used as the basis for the development of high polymorphic SSR primers. To achieve this goal, our study considered microsatellite markers with a standard size of 2–6 bp, excluding single nucleotides. In order to determine the microsatellite loci, MISA v2.1 software was used to focus on nucleotide microsatellites with a minimum number of repeats of five [24]. We used Primer 3 software to design primers for specific *S. involucrata* genomic sequences based on the reading parameters of the microsatellite region. The expected amplified fragment length ranged from 100 to 300 bp [25].

### 2.3. PCR Amplification and Electrophoresis Detection

Genomic DNA was extracted from *S. involucrata* plant materials using the DNAsecure plant kit (Tiangen Biotech (Beijing) Co., Ltd., Beijing, China) according to the manufacturer’s guidance. In total, 18 pairs of primers were randomly selected (Appendix A). The forward primers 5′ends were labeled with FAM blue, a fluorescent dye (Shanghai General Biotechnology, Shanghai, China), for easy scoring in genotyping. The selected primers were used for the PCR amplification of all sample DNA in a 25 μL reaction system, including 1 μL template DNA, 1 μL upstream and 1 μL downstream primers, 2 × EasyTaq PCR SuperMix 12.5 μL, and ddH2O 9.5 μL. The amplification reaction procedure consisted of three stages: The PCR reaction was carried out in a thermal cycler, with a single step denaturation at 93 °C for 3 min, followed by a denaturation at 94 °C for 30 s, followed by annealing at a temperature of 30 s according to the specific Tm value of each primer, then extending at 65 °C for 90 s, and finally extending at 65 °C for 5 min. The successful amplification was confirmed by 2% agarose gel electrophoresis. PCR products were detected by capillary electrophoresis and fluorescence labeling.

### 2.4. Data Analysis

The Hardy–Weinberg (HWE) equilibrium test was performed using GenAlEx 6.5 software. The linkage disequilibrium test was performed using GENEPOP on the Web (https://genepop.curtin.edu.au/genepop_op2.html (accessed on 2 April 2024)) [26]. The number of alleles (Na), effective number of alleles (Ne), Shannon’s information index (I), observed heterozygosity (Ho), expected heterozygosity (He), and other genetic diversity indexes were calculated by Gen Al Ex 6.5 software [27]. The polymorphic information content (PIC) was calculated by Power marker v3.25 software [28]. Nei’s genetic diversity index was calculated using Popgene 32 [29], and principal coordinate analysis (PCoA) was conducted using GenAlEx 6.5 software. The genetic distance (GD) matrix of 11 *S. involucrata* populations was calculated using the distance-based module in GenAlex 6.5 software. Then, molecular analysis of variance (AMOVA) was used to evaluate the contribution of genetic variation among and within populations. Bayesian assignment testing in STRUCTURE 2.3.4. software was used [30]. The parameters were set to K = 2–13, and each K value was run 20 times. The relevant parameters were set as follows: the length of the Burnin Period was 5 × 10^4^, the number of MCMC Reps after the Burnin was 5 × 10^5^, and the mixed model was used. For the numerical results corresponding to the K value of each run (stored in the Results folder), Structure Harvester was used to find the best ΔK value.

## 3. Results

### 3.1. Analysis of the Distribution of SSRs in the Genome of S. involucrata

Utilizing the MISA v2.1 software, we conducted a comprehensive screen of the 168.12 Mb genome of *S. involucrata*, uncovering a total of 673,244 SSR markers. These markers spanned a total length of 12,818,069 base pairs. The frequency and density of SSRs across the entire genome were calculated to be 4004.61 SSR/Mb and 76,244.85 bp/Mb, respectively. This represents a significant proportion of 7.62% of the genome sequence (Table 1).

The length of the SSR found in the whole genome of *S. involucrata* was 10~14,184 bp, with an average length of 19.04 bp. The most common repeat sequence length is 12 bp, with 108,391 occurrences, followed by 10 bp and 14 bp, with frequencies of 85,161 and 71,046 respectively (Figure 2).

The repeat motifs of each SSR locus were analyzed, and it was found that the number of repeats ranged from 5 to 3546. Most of the loci had 10 tandem repeats (17.07%), followed by loci with 6 tandem repeats (16.18%) (Figure 3).

In the whole genome of *S. involucrata*, dinucleotide repeats are the most common, followed by mononucleotide and trinucleotide repeats. The total length of the SSRs in the genome was 12,818,069 bp, and the total length of the SSRs containing one, two, three, four, five, and six nucleotide repeats was 2,264,331 bp, 8,661,590 bp, 1,338,753 bp, 360,580 bp, 61,925 bp, and 130,890 bp, respectively. The average length of each basic sequence was 12.29 bp, 21.34 bp, 18.70 bp, 59.48 bp, 27.62 bp, and 41.28 bp, respectively (Table 2). 

### 3.2. Linkage Disequilibrium Tests 

Linkage disequilibrium analysis was performed on the two sites, and the *p* value of the combination of 9 sites was significant (*p* < 0.05), which deviated from the balance (Table 3). Therefore, due to linkage disequilibrium, all further analysis removed S4, S31, S35, and other sites.

### 3.3. Genetic Diversity of the S. involucrata

A comprehensive analysis was conducted on 112 samples collected from 11 populations of *S. involucrata*, utilizing 15 SSR loci (Table 4). The results illuminated a total of 48 alleles across the loci, with an average of 3.182 alleles per locus, ranging from 2 to 7. Notably, the locus S10 exhibited the highest number of alleles. The average effective number of alleles (Ne) was 2.372, with the highest value of 5.700 observed at the S10 locus and the lowest of 1.452 observed at the S16 locus. The Shannon information index (I) varied from 0.376 to 1.818, while the observed heterozygosity (Ho) ranged from 0 to 1. The expected heterozygosity (He) values spanned from 0.234 to 0.811, and unbiased expected heterozygosity (uHe) ranged between 0.246 and 0.863. The fixation index (Fst) values were between 0.066 and 0.486, with an average of 0.199, indicating considerable genetic differentiation among the samples. The polymorphism information content (PIC) values ranged from 0.883 to 0.990, with an average of 0.971, suggesting that the selected primer pairs demonstrated substantial polymorphism. The Nei’s genetic diversity (Nei’s) values were between 0.234 and 0.811, with an average of 0.470.

The selected markers were 11 amplified populations of *S. involucrata*, the Na values of 2.467–3.800, and the Ne values of 1.940–2.841. The I value is between 0.599 and 0.950, the Ho value is between 0.341 and 0.431, the He value is between 0.345 and 0.506, the uHe value is between 0.364 and 0.538, and the F value is between −0.111 and 0.403. Hardy–Weinberg equilibrium analysis showed that there were loci deviating from Hardy–Weinberg equilibrium in all populations. Many loci deviated from HWE were found in population 8 (12 loci) (Table 5).

### 3.4. Genetic Relationship and Population Structure Analysis 

Principal coordinate analysis (PCoA) is used to provide a spatial representation of the relative genetic distance between individuals and to determine the consistency of population differentiation defined by cluster analysis. The PCoA of 112 *S. involucrata* materials is shown in Figure 4. The first principal coordinate and the second principal coordinate account for 4.18% and 2.29% of the total variation of the total coordinate respectively. The principal coordinate analysis (PCoA) showed that the individuals of various populations were mixed with each other, and the 11 populations of *S. involucrata* could not be clearly grouped. Most of the individuals in each population overlapped, but Populations 2 and 3 were far away from the other populations. Bayesian clustering results show that the provisional statistic ∆K shows that the maximum likelihood value of K = 4 (Appendix A). The results of STRUCTURE showed that there were different degrees of penetration in the population of *S. involucrata*, and there was an obvious genetic structure (Figure 5).

## 4. Discussion

### 4.1. Genetic Diversity of S. involucrata

The implementation of molecular markers has significantly propelled the fields of molecular ecology and population genetics. In our study, SSR markers were identified through the gene sequencing data of *S. involucrata*, culminating in the development of 18 SSR molecular markers. These markers were evaluated based on their Polymorphism Information Content (PIC), where a value greater than 0.5 indicates high polymorphism, a value between 0.25 and 0.5 suggests medium polymorphism, and a value less than 0.5 denotes low polymorphism [31]. The average PIC value of the 15 SSR markers in this study surpassed 0.5, underscoring their utility for analyzing the genetic diversity and population structure of *S. involucrata*.

Genetic variation is pivotal for local adaptation and the evolutionary process of species. Unraveling the genetic diversity of rare plants is essential for formulating robust long-term management and conservation strategies [32,33,34]. Our study revealed that *S. involucrata* possesses considerable genetic diversity (I = 0.837, He = 0.470), surpassing that of other endangered alpine flora such as *Eryngium alpinum* Lapeyr. (I = 0.283, Nei = 0.198) [35], *Isoetes hypsophila* Hand.-Mazz. (He = 0.039, Hs = 0.084, I = 0.061) [36], *Cerastium alpinum* L. (He = 0.085) [37], and *Sinadoxa corydalifolia* C. Y. Wu, Z. L. Wu & R. F. Huang (He = 0.368) [38]. The genetic diversity of plant species is influenced by a multitude of factors, including the distribution range, population size, life cycle, mating system, and gene flow [39]. Outcrossing plants, such as *S. involucrata*, typically exhibit higher genetic diversity and lower population differentiation compared to self-pollinating and clonal plants [40,41]. As a perennial and crosspollinated species, *S. involucrata* benefits from effective pollination mechanisms and long-distance seed dispersal, which contribute to its genetic polymorphism [8]. Due to the outcrossing system, it has high genetic diversity, which has also been reported in other Asteraceae plants [42,43,44]. SSR loci deviated from HWE in 11 populations of *S. involucrata*. In general, the population deviation from HWE may be due to the high heterozygosity or high invalid allele frequency at this locus [45]. The wild populations of *S. involucrata* were sporadically distributed, and the deviation of all populations from HWE may be caused by an insufficient population size and individual number. 

### 4.2. Genetic Differentiation of S. involucrata

Our analysis of molecular variation (AMOVA) among the 11 populations of *S. involucrata* indicated a higher degree of genetic differentiation within populations (97.440%), with a relatively minor contribution from genetic variation between populations (Table 6). This finding aligns with previous studies conducted in the Western Tianshan [16]. The breeding system is a determinant factor in the genetic variation observed within plant populations [46,47]. In plants that undergo crosspollination, the majority of genetic variation is distributed among individuals within a population, with a smaller proportion attributed to variation between populations [48]. *S. involucrata*’s floral biology, featuring male prematurity and hermaphroditism, along with its inflorescence and flower-opening mechanisms, promotes crosspollination [8].

Understanding gene flow in endangered plants is fundamental for their conservation and management. Moderate to high gene flow among populations is essential to preventing inbreeding depression and preserving genetic variation [49]. According to Wright, the gene flow (Nm) can be categorized into high (≥1.0), medium (0.250–0.99), and low (0.0–0.249) levels, with Nm values greater than 1 indicating significant gene flow between populations [50]. Our study demonstrated that *S. involucrata* exhibits substantial levels of gene flow mediated by pollen or seeds (mean Nm = 1.390 > 1). The sexual reproduction strategy of *S. involucrata* relies heavily on pollinators, particularly those belonging to the Bumblebeeidae family. Bumblebees, with their large body size and robust environmental adaptability, are capable of withstanding low temperatures and facilitating crosspollination over extended periods and distances. Additionally, the achene of *S. involucrata*, equipped with a long pappus, can be dispersed by wind, while the fruit can be spread through surface runoff. These mechanisms, along with the pappus’s ability to retain water and adhere to other organisms, contribute to the species’ diverse dispersal methods, primarily windborne with elements of water and animal-borne dissemination [8]. In order to clarify the genetic relationship of 112 accessions of *S. involucrata* from 11 populations, we used two different clustering methods. Using a two-dimensional PCoA distance matrix to visualize the relationship between samples makes it difficult to group and analyze them. In contrast, the cluster analysis using STRUCTURE showed a clear genetic structure between populations. *S. involucrata* relies on wind to spread seeds [8]. This mode of transmission may promote long-distance gene exchange. Wind can carry seeds to places far away from the mother plant, increasing the mating opportunities with other *S. involucrata* individuals. Therefore, different populations have frequent gene exchanges. The strong gene exchange of wind-borne plants has also been reported in other studies [51,52]. Geographical distance is not the only explanatory factor for the genetic differentiation of the *S. involucrata* population. The overall topography of the sampling site is high in the northwest and low in the southeast [53], the climate between the mountains is more complex, and the wind direction is more changeable. There are no mountains in the middle area of the sampling site that affect gene exchange. The altitude of the three sampling sites of the *S. involucrata* population is lower than that of other eastern sampling sites, and the western *S. involucrata* population has a stronger influence on it. Different terrains may form different niches, prompting species to adapt to specific environmental conditions, thus affecting the genetic structure [54]. Terrain differences can lead to changes in the microclimate, such as temperature, humidity, and wind speed, which may affect the survival and reproduction of organisms [55]. *S. involucrata* Population 1 is located upstream of the wind direction in this area, and the mountains block the gene exchange with *S. involucrata* Population 2. A more in-depth investigation of *S. involucrata* outside the study area can further explore the reasons for the special genetic structure.

### 4.3. Conservation of S. involucrata

Preserving genetic variation is a cornerstone of conservation efforts for endangered and threatened species [56]. Insights into genetic variation among and within populations are vital for crafting effective conservation management strategies [57]. Given the unique pharmacological properties of *S. involucrata* and its burgeoning applications in medicine, skincare, and health care [58], the overexploitation of this species has led to a precipitous decline in its natural reserves, ecological devastation, and challenges in cultivation [59]. Despite a ban on wild harvesting in China, the loss of wild *S. involucrata* in Xinjiang is estimated at approximately 40 tons annually [60]. Climate change, specifically the rising snowline in the Tianshan Mountains, has drastically reduced the habitable range for *S. involucrata*, making it increasingly rare below the 3400 m elevation [60]. The constriction of populations increases the risk of losing genetic polymorphism due to genetic drift and inbreeding. Urgent conservation measures are warranted to ensure the long-term survival of this rare species. Considering the scarcity of wild populations and the minimal genetic differentiation among them, it is imperative to protect all populations in the Bayinbuluke area. Enhancing in situ conservation efforts can help maintain current population sizes, establish protected areas for *S. involucrata*, and bolster its adaptability. To safeguard this precious medicinal resource, the Xinjiang Uygur Autonomous Region government has prohibited mountain mining and mandated a three-year restoration period postharvest [61]. Considering *S. involucrata*’s sole mode of reproduction being seed-based, its naturally low germination and survival rates—approximately 3% under natural conditions—and its 5–6-year maturation period from germination to flowering and fruiting [62], tissue culture techniques can be employed to expedite plant propagation and address the shortfall in natural resources.

## 5. Conclusions

The genetic analysis of *S. involucrata* underscores the importance of conserving genetic diversity and implementing targeted protection strategies. Our findings highlight the species’ high genetic diversity and the significant gene flow among populations, which are crucial for its evolutionary potential and adaptability. The conservation measures proposed are essential for maintaining the species’ resilience and ensuring its survival in the face of environmental challenges and anthropogenic pressures. The development and application of SSR markers offer a valuable tool for future research and conservation efforts, providing a foundation for evidence-based management strategies.

## Figures and Tables

**Figure 1 biomolecules-14-01010-f001:**
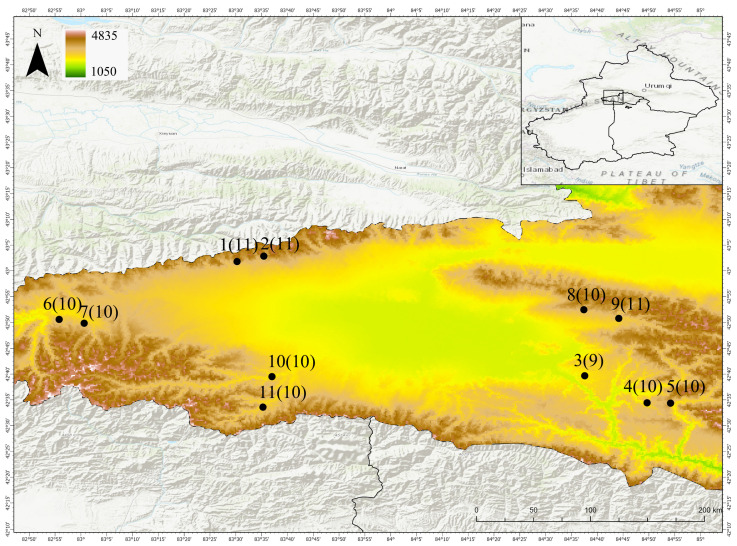
The sampling distribution map of 11 populations of *S. involucrata*. The solid circle represents the position of the sampling population, and the numbers in brackets represent the individuals of the population.

**Figure 2 biomolecules-14-01010-f002:**
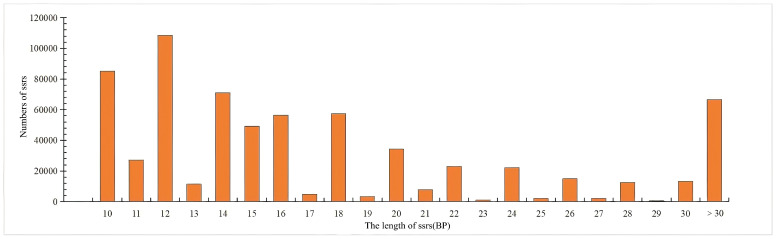
The length distribution of SSRs in the *S. involucrata* genome.

**Figure 3 biomolecules-14-01010-f003:**
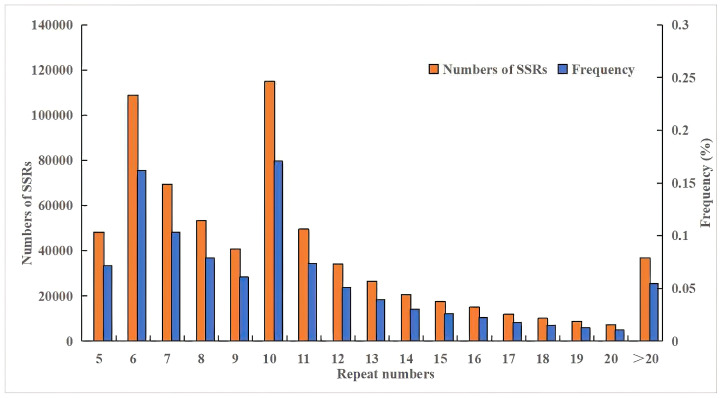
The distribution of the number and frequency of SSRs with different repetitions in the genome of *S. involucrata*.

**Figure 4 biomolecules-14-01010-f004:**
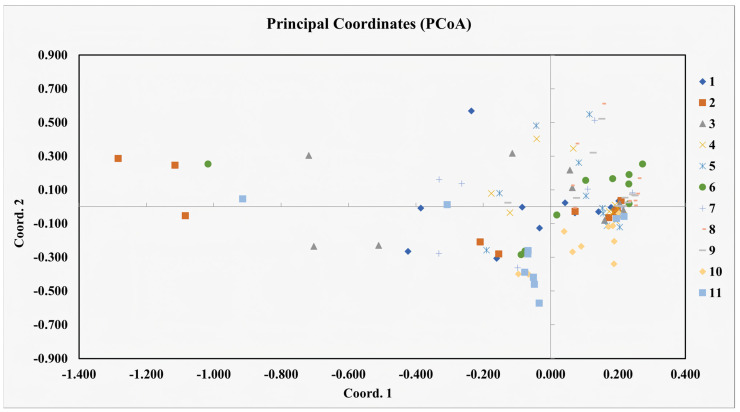
Principal coordinate analysis (PCoA) of 112 *S. involucrata* based on 15 SSR markers. The different colors and shapes represent different study populations. The first and second axes explained 4.18% and 2.29% of the genetic similarities among populations, respectively.

**Figure 5 biomolecules-14-01010-f005:**
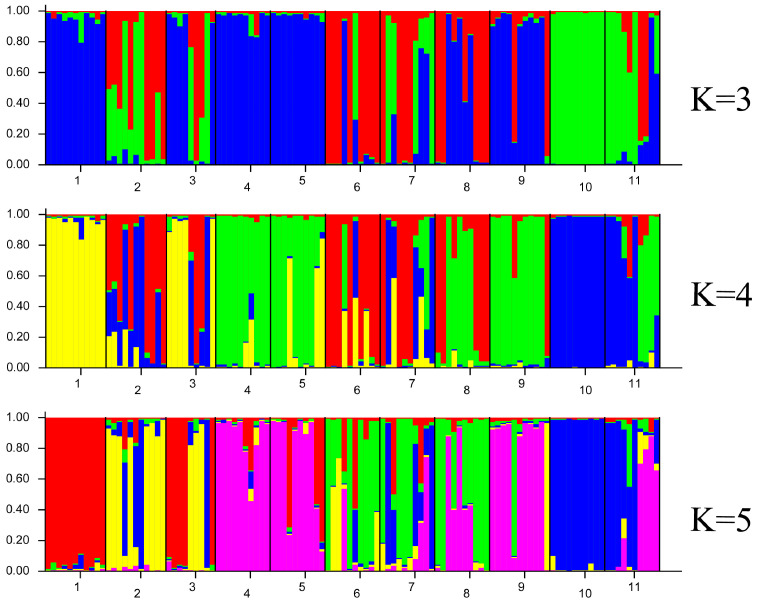
K = 3 to K = 5 cluster STRUCTURE ancestral proportion bar chart. Each individual is represented as a line segment, divided vertically by different colors, representing the proportion of ancestors estimated by the individual in each cluster. The number below the figure represents the sampling population.

**Table 1 biomolecules-14-01010-t001:** The result of SSRs loci in the *S. involucrata* genome.

Items	Numbers
Total size of the genome (Mb)	168.12
Total number of identified SSRs	673,244
Total length of SSRs (bp)	12,818,069
Frequency (SSRs/Mb)	4004.61
Density (bp/Mb)	76,244.85
Total content of genome SSRs (%)	7.62

**Table 2 biomolecules-14-01010-t002:** The main repeat motifs, number, frequency, proportion, and length of each nucleotide in the genome of *S. involucrata*.

Repeat Type	Predominant Type	Number	Proportion (%)	Frequency (SSRs/Mb)	Total Length (bp)	Average Length (bp)
Mono	A/T	184,196	27.36	1095.64	2,264,331	12.29
Di	AT/AT	405,972	60.30	2414.82	8,661,590	21.34
Tri	ATC/ATG	71,601	10.64	425.90	1,338,753	18.70
Tetra	ACAT/ATGT	6062	0.90	36.06	360,580	59.48
Penta	AACCC/GGGTT	2242	0.33	13.34	61,925	27.62
Hexa	AAGGAG/CCTTCT	3171	0.47	18.86	130,890	41.28
Total		673,244	100	4004.61	12,818,069	19.04

**Table 3 biomolecules-14-01010-t003:** The linkage disequilibrium test between two loci of eighteen polymorphic microsatellite loci in *S. involucrata*.

	S4	S10	S11	S15	S16	S20	S23	S24	S25	S26	S29	S30	S31	S32	S35	S36	S37	S38
S4																		
S10	0.474																	
S11	0.977	0.981																
S15	0.414	0.457	0.966															
S16	0.370	0.819	0.970	0.139														
S20	0.024 *	0.487	0.994	0.911	0.210													
S23	0.906	0.373	1.000	0.848	0.950	0.558												
S24	0.637	0.388	1.000	0.727	0.875	0.495	0.841											
S25	0.060	0.529	0.788	0.618	0.775	0.110	0.131	0.083										
S26	0.315	0.506	0.703	0.922	0.289	0.382	0.917	0.979	0.057									
S29	0.995	0.667	0.992	0.925	0.913	0.999	0.864	0.999	0.908	0.869								
S30	0.297	0.837	0.837	0.232	0.937	1.000	0.999	0.883	0.166	0.130	0.971							
S31	0.277	0.393	0.979	0.520	1.000	0.744	0.694	0.565	0.026 *	0.390	0.996	0.198						
S32	0.875	0.503	1.000	0.812	0.384	0.940	0.825	0.249	0.111	0.950	0.649	0.351	0.016 *					
S35	0.013 *	0.940	1.000	0.431	0.423	0.202	0.351	0.169	0.001 *	0.332	0.679	0.054	0.018 *	0.016 *				
S36	0.963	0.186	0.987	0.567	0.995	0.954	0.548	1.000	0.385	0.998	1.000	0.557	0.836	0.976	0.404			
S37	0.782	0.722	1.000	0.405	0.648	0.934	0.535	0.782	0.250	0.926	0.941	0.711	0.003 *	0.079	0.004 *	0.642		
S38	0.978	0.161	0.990	0.727	0.991	0.844	0.706	0.433	0.997	0.774	0.893	0.826	0.597	0.923	0.175	1.000	0.956	

*: significant level at *p* < 0.05.

**Table 4 biomolecules-14-01010-t004:** Statistical values of microsatellite markers in 112 samples of 11 *S. involucrata* populations.

Locus	Na	Ne	I	Nm	Ho	He	uHe	Fst	PIC	Nei’s
S10	7	5.700	1.818	1.664	0.702	0.811	0.863	0.131	0.971	0.811
S11	3	1.901	0.754	1.120	0.008	0.438	0.461	0.182	0.978	0.438
S15	6	3.812	1.437	3.552	1.000	0.697	0.737	0.066	0.984	0.697
S16	2	1.452	0.376	0.522	0.000	0.234	0.246	0.324	0.980	0.234
S20	4	2.736	1.011	1.255	0.125	0.549	0.580	0.166	0.983	0.549
S23	2	1.639	0.529	1.191	0.000	0.345	0.366	0.173	0.971	0.345
S24	3	2.028	0.804	0.750	0.151	0.455	0.482	0.250	0.968	0.455
S25	3	2.168	0.796	0.998	0.498	0.456	0.482	0.200	0.982	0.456
S26	2	1.645	0.510	0.583	0.170	0.325	0.359	0.300	0.883	0.325
S29	3	2.507	1.026	1.233	0.709	0.592	0.631	0.169	0.955	0.592
S30	3	2.643	1.032	2.570	0.991	0.607	0.639	0.089	0.990	0.607
S32	3	2.289	0.848	1.923	1.000	0.553	0.585	0.115	0.983	0.553
S36	2	1.618	0.552	2.444	0.000	0.342	0.360	0.093	0.982	0.342
S37	2	1.478	0.430	0.264	0.193	0.277	0.293	0.486	0.977	0.277
S38	2	1.964	0.631	0.774	0.350	0.368	0.389	0.244	0.978	0.368

Note: Na = number of alleles; Ne = effective alleles; I = Shannon information index; Nm = gene flow; Ho = observed heterozygosity; He = expected heterozygosity; uHe = unbiased expected heterozygosity; Fst = genetic differentiation coefficient; PIC = polymorphism information content; Nei’s = Nei’s genetic diversity.

**Table 5 biomolecules-14-01010-t005:** Summary of genetic statistics for *S. involucrata* at population level.

Pop		N	Na	Ne	I	Ho	He	uHe	F	Percentage of Deviation from HWE Site (%)
1	Mean	9.867	3.267	2.490	0.854	0.374	0.480	0.506	0.333	66.667
	SE	0.307	0.672	0.473	0.140	0.116	0.058	0.061	0.192	
2	Mean	8.400	3.200	2.458	0.848	0.426	0.475	0.506	0.193	53.333
	SE	0.335	0.518	0.424	0.137	0.110	0.060	0.064	0.193	
3	Mean	7.667	3.800	2.651	0.950	0.403	0.501	0.538	0.326	53.333
	SE	0.333	0.611	0.475	0.144	0.114	0.058	0.063	0.177	
4	Mean	9.467	3.067	2.340	0.868	0.431	0.501	0.529	0.121	60.000
	SE	0.215	0.371	0.225	0.117	0.120	0.059	0.063	0.212	
5	Mean	9.333	3.067	2.350	0.864	0.420	0.496	0.525	0.167	46.667
	SE	0.211	0.358	0.251	0.117	0.108	0.061	0.064	0.181	
6	Mean	8.933	3.400	2.841	0.915	0.375	0.506	0.536	0.289	73.333
	SE	0.206	0.668	0.570	0.160	0.112	0.068	0.072	0.205	
7	Mean	9.133	3.267	2.234	0.847	0.375	0.480	0.509	0.299	73.333
	SE	0.376	0.483	0.279	0.112	0.107	0.048	0.051	0.185	
8	Mean	9.733	3.133	2.428	0.865	0.346	0.499	0.526	0.372	80.000
	SE	0.153	0.487	0.319	0.126	0.116	0.055	0.059	0.212	
9	Mean	10.667	3.467	2.283	0.849	0.415	0.453	0.476	0.274	66.667
	SE	0.159	0.424	0.292	0.130	0.107	0.066	0.069	0.167	
10	Mean	9.667	2.467	1.940	0.599	0.418	0.345	0.364	−0.111	53.333
	SE	0.187	0.435	0.301	0.143	0.118	0.075	0.079	0.182	
11	Mean	9.867	3.267	2.490	0.854	0.374	0.480	0.465	0.403	66.667
	SE	0.307	0.672	0.473	0.140	0.116	0.058	0.062	0.186	

Note: N = number of individuals sampled; Na = number of alleles; Ne = effective alleles; I = Shannon information index; Ho = observed heterozygosity; He = expected heterozygosity; uHe = unbiased expected heterozygosity; F = fixation index; percentage of deviation from HWE loci = (number of loci significantly deviated from HWE/total number of loci) × 100%; SE = standard error of the mean.

**Table 6 biomolecules-14-01010-t006:** Analysis of molecular variance (AMOVA) of genetic variation within and among groups of *S. involucrata*.

Source of Variation	df	SS	Est. Var.	Variation (%)
Among Pops	10	297.870	0.617	2.560
Within Pops	101	2373.701	23.502	97.440
Total	111	2671.571	24.119	100.000

## Data Availability

Data will be made available on request.

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
