# Peer review of "Development of SSR Markers and Evaluation of Genetic Diversity of Endangered Plant Saussurea involucrata"

_biomolecules, 2024, doi:10.3390/biom14081010_

Round 1

Reviewer 1 Report

Comments and Suggestions for Authors

I have reviewed the manuscript titled „Development of SSR markers and evaluation of genetic diversity of endangered plant Saussurea involucrata“. The manuscript is structured well and is easily readable. I commend the authors on developing SSR marker system for a species that is considered vulnerable due to overexploitation.

I do have some remarks that need to be addressed before the manuscript be suitable for publishing. I have put all my comments in the manuscript file that I'm attaching, but here are the major ones:

·       It seems that hyphens have remained in the text from the prior version of the manuscript. I have highlighted several instances, but please go through text carefully and correct this error.

·       At first mention of all species in the manuscript, please use full name, including the author, according to botanical nomenclature.

·       I highly suggest reconsidering Table 1. Information provided can be added elsewhere (population serial number and number of sampled individuals can be added to one of the results table), while coordinates (especially of this resolution and for VU species) are not advisable to reveal. Instead I suggest constructing a map of sampled locations (potentially add polygons denoting species overall distribution) in order to introduce the reader with the region and the geographical scope of our research.

·       Nei's genetic diversity index was mentioned in the Materials and Methods section as something that was calculated, yet there is no results based on this index. Was the index used in PCoA or was is omitted from the final manuscript version? Please revise.

·       Figures are of low quality. If accepted, please provide a higher resolution figures

·       The subsection 3.3. Development of SSR markers in S. involucrata genome is not about the development of markers, rather is a descriptive statistic on loci and population level. I suggest renaming it to reflect the content

·       In Table 5 you present the results of testing HWE. Why did you test the loci for deviation from HWE, but not populations? This research is not just about development of SSRs but also a study on population diversity of the species. It would be more beneficial to test HWE for populations rather than just loci.

·       On lines 244-246 you mention that the first two axes account for a very small variation. On the plot depicting these two axes you get a very crowded and overlapping samples. I would suggest a Bayesian analysis of population structure instead of PCoA or a combination of both methods. based on PCoA you can see there potentially exists some kind of structure, but it needs to be tested using a proper method. Overlaying the results of Structure analysis onto geographical space may reveal some geographical or ecological clustering. If Structure analysis reveals no genetic structure, then your conclusions are sound.

Comments on the Quality of English Language

Minor corrections to the text are needed ut overall the English is fairly good.

Author Response

  1. It seems that hyphens have remained in the text from the prior version of the manuscript. I have highlighted several instances, but please go through text carefully and correct this error.

An: Thank you for your reminder. we have deleted the wrong hyphen in the article, and read the article carefully to correct the relevant errors.

  1. At first mention of all species in the manuscript, please use full name, including the author, according to botanical nomenclature.

An: It is very important to use the botanical nomenclature including the author for the first mentioned species. The species first mentioned in the manuscript have been corrected according to the botanical nomenclature.See them in Line 42, p 1. Line 230-232, p 9.

  1. I highly suggest reconsidering Table 1. Information provided can be added elsewhere (population serial number and number of sampled individuals can be added to one of the results table), while coordinates (especially of this resolution and for VU species) are not advisable to reveal. Instead I suggest constructing a map of sampled locations (potentially add polygons denoting species overall distribution) in order to introduce the reader with the region and the geographical scope of our research.

An: The sampling information in Table 1 has been represented by a map.

See them in Fig 1.

  1. Nei's genetic diversity index was mentioned in the Materials and Methods section as something that was calculated, yet there is no results based on this index. Was the index used in PCoA or was is omitted from the final manuscript version? Please revise.

An: Thanks for your careful checks. We are sorry for our carelessness. The results of Nei 's genetic diversity have been added.See them in Table 4.

  1. Figures are of low quality. If accepted, please provide a higher resolution figures.

An: We 've made changes to the picture to make it more readable. See them in Fig 2, Fig 3, Fig 4.

  1. The subsection 3.3. Development of SSR markers in S. involucrata genome is not about the development of markers, rather is a descriptive statistic on loci and population level. I suggest renaming it to reflect the content

An: We have corrected it to Genetic diversity of the S. involucrata.

  1. In Table 5 you present the results of testing HWE. Why did you test the loci for deviation from HWE, but not populations? This research is not just about development of SSRs but also a study on population diversity of the species. It would be more beneficial to test HWE for populations rather than just loci.

An: We feel great thanks for your professional review work on our article. We have performed the HWE test on the S. involucrata populations. See them in Table 5.

  1. On lines 244-246 you mention that the first two axes account for a very small variation. On the plot depicting these two axes you get a very crowded and overlapping samples. I would suggest a Bayesian analysis of population structure instead of PCoA or a combination of both methods. based on PCoA you can see there potentially exists some kind of structure, but it needs to be tested using a proper method. Overlaying the results of Structure analysis onto geographical space may reveal some geographical or ecological clustering. If Structure analysis reveals no genetic structure, then your conclusions are sound.

An: We have added the results of Bayesian analysis of population structure to the annex, which can effectively support the results of PCoA.

Reviewer 2 Report

Comments and Suggestions for Authors

Indications for improvements are shown as baloon notes in the MS file.

The authors should pay particular attention to the legends of Figures and Tables. Actually the legends should be re-written anew.

Author Response

We are very grateful to you for your professional review of this article, which has effectively improved the quality of the manuscript. We appreciate your suggestions regarding the article writing and have added them accordingly in the manuscript, with the following detailed revisions.

  1. Indications for improvements are shown as baloon notes in the MS file.

An: 1. The Country and regional information has been added. See them in line 22, p 1.

  1. The information of Saussurea involucrata plant family has been added.See them in line 42, p 1.
  1. line 61, p 2.

An: Saussurea involucrata belongs to the national second-level protected plant category in the ' National Key Protected Wild Plants List ' approved by the State Council on August 7,2021.

  1. line 74, p 2.

An: The references of this part have been changed to 2 articles.

  1. Table 1, p 2.

An: The information in Table 1 has been changed to a map representation, and Number represents the sampling individual of the population.

  1. line 110, p 3.

An: The content of the data analysis has been revised and the analysis software used in the article has been described in more detail.

  1. Table 6, p 6.

An: We were really sorry for our careless mistakes. Thank you for your reminder. The Table 6 has been modified.

  1. Fig. 3, p 7. populations?

An: And what the 15 markers did? The Discussion section does not say enough for their ordination.

The numbers 1-11 represent 11 populations of Saussurea involucrata. The results of PCoA were analyzed based on 15 SSR molecular markers, and the discussion section has been supplemented.

  1. The authors should pay particular attention to the legends of Figures and Tables. Actually the legends should be re-written anew.

An: Figures and table legends have been rewritten.

Reviewer 3 Report

Comments and Suggestions for Authors

All comments are in the attached file.

Author Response

We are very grateful to you for your professional review of this article, which has effectively improved the quality of the manuscript. We appreciate your suggestions regarding the article writing and have added them accordingly in the manuscript, with the following detailed revisions.

  1. The article is well structured, and its content is well explained. Therefore, it could be published in this version although some small improvements would be advisable. In general, remember that all Tables and Figures must be understandable per se without the need to refer to the text. Titles should be more explicit, both headers and column names, and all necessary table/figure captions should be added.

An: We have modified the titles of tables and pictures and added all necessary Tables / Figures descriptions. See them in Fig 1, Fig 2, Fig 3, Fig 4, Table 1, Table 2, Table 3, Table 4, Table 5, Table 6.

  1. The distance between the chosen populations is not established, nor is it specified why they have been chosen. Nor are the sampling conditions indicated, what characteristics the chosen individuals should or should not have: for example, health, representativeness of the population, or any other outstanding trait.

An: The sampling information has been expressed in the form of figure, and the distance between different groups is easier to understand, and the sampling conditions are supplemented in the article. Eleven wild populations of Saussurea involucrata were collected, and each population was at least 7 km apart. In each population, we randomly selected healthy individuals of Saussurea involucrata (each individual is at least 50 meters apart). See them in Fig 1.

Table 1: Replace with a map of the location of the area where it will be clear which area of China is being discussed and how the sampling was distributed. All the information in Table 1 should be contained in the graph in a way that is clearer to the reader. Perhaps the serial number is not necessary, but the name of the populations studied is.

An: The sampling information in Table 1 has been replaced by a map, which clarifies the area of discussion and describes the sampling information in more detail. Since the sampling populations are distributed in the same area, we use the number to distinguish different sampling populations.

Table 2: The title is too general, specify more, put overview does not specify anything about the content of this table.

An: The title of Table 2 has been modified.

Figures 1 & 2:The same comments apply to Figures 1 and 2. Make the graphs in colour, it is more visual. Where can you see in the Figures what is said in the text about the most frequent SSRs? Increase the font size of the axes, the indications in Figure 1 are not legible.

An: Figure 1 and Figure 2 have been changed to color Figures to make them more visible. The most frequent SSR statement in the text has been modified, and all images have been replaced with higher quality.

  1. Table 3: The same comment, the title is not explanatory of the content.

An: The title of Table 3 has been modified.

Table 4: You refer to the identification of the SSRs chosen for the first time just before the Table, and the Table itself does not identify what the axes are or what type of analysis has been performed. Again, there is a lack of information and precision for the Table to be explanatory per se.

An: The analysis types in Table 4 have been added to the materials and methods, and the title of Table 4 has been modified.

Table 5: If the number of samples was always the same, this should not be indicated in the Table but in the Table title. Significance levels should be indicated at the foot of the Table.

An: Thanks for your suggestion, The number of samples has been marked in Table 5 and the significance level is marked at the foot of the table.

  1. Table 6 has no title. All the necessary information must be added at the foot of the table. This table is about to be completed.

An: Thanks for your careful checks. We are sorry for our carelessness. The title of Table 6 has been added, and all necessary information has been added at the foot of the table.

  1. Figure 3: Find a way to make the results less blurred, perhaps simply a larger representation in size. If we had a distribution of the populations as indicated in Table 1, it would be clearer to see whether the populations are geographically close or far away and could perhaps lead to some further commentary.

An: We sincerely appreciate the valuable comments. Figure 3 has been replaced with a higher-quality figure, and the information in Table 1 is represented by a map, which helps to understand the distribution of the sampled population more easily.

  1. Table 7: Adding information as already mentioned above.

An: The information in Table 7 has been added.

  1. Others:

Keep in mind that in the text, from paragraph 3.3 onwards, hyphens appear in the middle of

words where they should not appear. Examples: Lines 158,163,166,183,186, 220, 221, 224,

230, etc.

An: We have deleted the wrong hyphen in the article, and read the article carefully to correct the relevant errors.

Round 2

Reviewer 1 Report

Comments and Suggestions for Authors

Dear authors,

thank you for accepting the feedback and performing additional analysis of your data. A few additional comments after reviewing your second version of the manuscript:

1. You have added to the table 5 loci that show deviation from HWE for each of the populations. It is more common to provide an average deviation across all loci (this will both reduce the size of the table and give the readers information about which populations exhibit deviation from HWE, which you then can discuss in combination with other results.

2. Thank you for performing the Bayesian analysis of genetic structure on your data. I must correct you when stating that the analysis confirms the results of the PCOA in that there is no structure in your populations. After examining the supplement file and comparing it to the map of sampled populations it is obvious that there is some structure present; namely populations 4,5,8 and 9 (eastern populations) are dominated by a blue gene pool; additionally, populations 10 and 11 have separated in a yellow gene pool and population 1 is predominately in a green gene pool. NW populations (6,7,2) are hybrid populations for red and yellow gene pools.

My suggestion is to replace the PCoA graph with the results of Structure for K=4 (or add this graph to the main text) and discuss the possible reasons for this distribution in population structure. Is population 1 somehow isolated from the rest of the species distribution? How about population 3 which is geographically close to the eastern populations, but shows a different structure? etc.

3. Please provide more information on the methodology for Structure analysis (Burn-in period, MCHC, number of repetitions) to increase the replicability of your study. Similarly, provide us the information on a number of permutations for the AMOVA.

Author Response

We are very grateful to you for your professional review of this article, which has effectively improved the quality of the manuscript. We appreciate your suggestions regarding the article writing and have added them accordingly in the manuscript, with the following detailed revisions.

  1. You have added to the table 5 loci that show deviation from HWE for each of the populations. It is more common to provide an average deviation across all loci (this will both reduce the size of the table and give the readers information about which populations exhibit deviation from HWE, which you then can discuss in combination with other results.

An: Thanks for your suggestion. An average deviation across all loc has been added to Table 5, expressed as a percentage of loci deviating from HWE (Percentage of deviation from HWE loci = (number of loci significantly deviated from HWE / total number of loci) × 100 %), which reduces the size of the table and makes it easier to read, and the discussion of this section has been added. See them in Line 251-255, p 10.

  1. Thank you for performing the Bayesian analysis of genetic structure on your data. I must correct you when stating that the analysis confirms the results of the PCOA in that there is no structure in your populations. After examining the supplement file and comparing it to the map of sampled populations it is obvious that there is some structure present; namely populations 4,5,8 and 9 (eastern populations) are dominated by a blue gene pool; additionally, populations 10 and 11 have separated in a yellow gene pool and population 1 is predominately in a green gene pool. NW populations (6,7,2) are hybrid populations for red and yellow gene pools.

My suggestion is to replace the PCoA graph with the results of Structure for K=4 (or add this graph to the main text) and discuss the possible reasons for this distribution in population structure. Is population 1 somehow isolated from the rest of the species distribution? How about population 3 which is geographically close to the eastern populations, but shows a different structure? Etc.

An: We think this is a constructive suggestion. The results of Bayesian analysis of genetic structure showed that there was a genetic structure in S. involucrata. We modified this part of the article and added the results of genetic structure from K = 3 to K = 5 to the article. In the discussion section of 4.2, the reasons for the genetic structure of S. involucrata and the reasons for the special genetic structure of population 1 and population 3 were analyzed. See lines 283-306, p 11.

  1. Please provide more information on the methodology for Structure analysis (Burn-in period, MCHC, number of repetitions) to increase the replicability of your study. Similarly, provide us the information on a number of permutations for the AMOVA.

An: Specific parameters for the genetic structure analysis methods and specific analysis methods for AMOVA have been added to the 2.4. Data Analysis section. See them in Line 124-133, p 4.